# SALL4: An Intriguing Therapeutic Target in Cancer Treatment

**DOI:** 10.3390/cells11162601

**Published:** 2022-08-20

**Authors:** Shiva Moein, Daniel G. Tenen, Giovanni Amabile, Li Chai

**Affiliations:** 1Cancer Science Institute of Singapore, Singapore 117599, Singapore; 2Harvard Stem Cells Institute, Harvard Medical School, Boston, MA 02115, USA; 3Believer Pharmaceuticals, Inc., Wilmington, DE 19801, USA; 4Department of Pathology, Brigham & Women’s Hospital, Harvard Medical School, Boston, MA 02115, USA

**Keywords:** SALL4, neoplasm, drug development, molecular glue

## Abstract

Spalt-Like Transcription Factor 4 (SALL4) is a critical factor for self-renewal ability and pluripotency of stem cells. On the other hand, various reports show tight relation of SALL4 to cancer occurrence and metastasis. SALL4 exerts its effects not only by inducing gene expression but also repressing a large cluster of genes through interaction with various epigenetic modifiers. Due to high expression of SALL4 in cancer cells and its silence in almost all adult tissues, it is an ideal target for cancer therapy. However, targeting SALL4 meets various challenges. SALL4 is a transcription factor and designing appropriate drug to inhibit this intra-nucleus component is challenging. On the other hand, due to lack of our knowledge on structure of the protein and the suitable active sites, it becomes more difficult to reach the appropriate drugs against SALL4. In this review, we have focused on approaches applied yet to target this oncogene and discuss the potential of degrader systems as new therapeutics to target oncogenes.

## 1. Introduction

Spalt-Like Transcription Factor 4 (SALL4), a member of the SALL family, is a regulator of embryonic stem cells that plays a crucial role in cell renewal and proliferation. SALL4 expression starts as soon as two-cell stage. In the blastocyst SALL4 over expression is observed in the inner cell mass (ICM) and the trophectoderm and later in midbrain, genital tubercle, limb, and tail buds [1]. Knock down studies show its prominent role in different stages of fetal development [2] and mutation in the SALL4 gene is the cause of an autosomal dominant syndrome named Okihiro that mainly affects eyes, bones of arms and hands, heart, and renal system [3]. However, the gene is silenced in most of the adult tissues except germ cells and CD34+ hematopoietic stem cells [4,5].

Reactivation of SALL4 in cancerous cells is a key event in tumorigenesis and confers malignancy features to cells. A meta-analysis that assessed the risk of SALL4 activation with all-cause mortality and recurrence resulted in hazard ratios (HR) of 1.4 (95% confidence interval: 1.19–1.65) and 1.52 (95% confidence interval: 1.22–1.89), respectively. These data indicate that SALL4 is associated with a poor survival rate and suggests it as a potential biomarker of cancer prognosis [6].

SALL4 has two isoforms. SALL4A originates from full length transcript, while SALL4B as a spliced isoform lacks a part of exon 2. The SALL4 protein contains several zinc finger cluster (ZFC) domains spread in different parts of the protein which give it both transcription activation and repression functions [7,8]. SALL4A has one C2HC domain at N-terminal, one double C2H2 (ZFC4) at C-terminal and one double C2H2 (ZFC2) and one triple C2H2 (ZFC3) at the middle of the protein. However, SALL4B lacks the middle zinc finger domains ZFC2 and ZFC3 [8]. Localization of the protein to heterochromatin structure is mediated by ZFC4. Besides ZF domains, SALL4 contains a glutamine (Q)-rich region at N-terminal of the protein which is necessary for formation of dimer structures with SALL family members [9]. As a transcription regulatory factor, SALL4 has a nuclear localization signal at amino acids 64–67 [10]. 

SALL4 abundancy is regulated by transcriptional and post-transcriptional mechanisms. It is shown that Signal Transducer and Activator of Transcription 3 (STAT3) and caudal type homeobox 1 (CDX1) promote SALL4 expression in breast and gastric cancers, respectively [11,12]. Moreover, SALL4 participates in an interconnected auto-tuning circuit with Octamer-binding transcription factor 4 (Oct4), (SRY-box 2) Sox2, and Nanog, regulating not only each other expression but also their own expression in mouse embryonic stem cells [13]. WNT signaling also modulates SALL4 expression through T-cell factor/lymphoid enhancer factor (TCF/LEF) sequence in the promoter of the gene [14]. The epigenetic modifications also affect SALL4 expression through the CpG island located at the Exon 1/Intron 1 region and reported to be hypomethylated both in embryonic stem cell and cancer cells [15]. De-methylation of this region can lead to SALL4 activation [16,17,18].

SALL4 is a dual-mode transcription factor that can activate and repress clusters of genes that are critical for self-renewal and pluripotency. SALL4 performs its repressive role through interaction with Nucleosome Remodeling and Deacetylase complex (NuRD). Phosphatase and tensin homolog (PTEN) that is an important tumor suppressor gene and blocks Phosphatidylinositol-3-Kinase/Protein Kinase B signaling (PI3K/AKT) is a downstream target of SALL4 [19,20]. Lysine Demethylase 3A (KDM3A), Forkhead-Box (FOX), B-cell lymphoma (BCL), and Krüppel-like factor (KLF) are the other genes that are repressed by SALL4 leading to promotion of tumor progression in cancer cells [21].

The transcription activation role of SALL4 is also reported for Homeobox A9 (HOXA9) through interaction with mixed-lineage leukemia (MLL) protein. Moreover, SALL4 directly binds to c-MYC promoter and enhances its expression. The oxidative phosphorylation genes are the other targets of SALL4 protein [22]. SALL4 binding sites in mouse embryonic stem cells was found by a genome-wide analysis using ChIP-on-chip assays. 3223 genes were determined as potential target genes of SALL4 among which genes related to Nuclear Factor Kappa-light-chain-enhancer of activated B cells (NFKB), apoptosis, Platelet-Derived Growth Factor (PDGF), P53, Transforming Growth Factor Beta (TGFB), and Wnt/B-catenin were enriched [23].

Considering the critical role of SALL4 in cancer cell survival besides the difference in the expression of SALL4 in healthy and cancer cells, this protein seems to be a key target for cancer treatment. Hence, identification of novel drugs targeting this oncogene may be critical for establishing new effective treatments for intractable tumors. Here we reviewed the major molecular mechanisms associated with SALL4 in different cancers and also extensively went through the drugs designed against SALL4 and its interactions.

## 2. Molecular Mechanisms Leading to Aberrant Activation of SALL4 in Cancer

SALL4 expression is regulated by various molecular mechanisms involving transcriptional, post-transcriptional, and epigenomic levels. Different transcription factors regulate expression of SALL4. STAT3 is one of the main regulators of SALL4 in breast cancer and hepatocellular carcinoma (HCC) [11]. It is also shown that SALL4 is directly targeted by caudal-related homeobox 1 (CDX1) protein in gastric epithelial cells [12]. Moreover, there is an interconnected regulatory loop constituted of SALL4, OCT4, NANOG, and SOX2 that each transcription factor regulates both its own expression as well as the other genes in the loop [13,24]. Interestingly, the increased levels of SALL4 expression are controlled through a self-repressive circuit. Moreover, Du et al. showed that the expression of SALL4 is induced by Epidermal Growth Factor Receptor (EGFR) through activation of the ERK1/2 pathway [25]. Activation of SALL4 in EGFR mutated lung cancer cells resulted in spheroid formation and expression of stem cell factor CD44. Wnt/β-catenin is another prominent regulator of SALL4 expression. The SALL4 promoter and intergenic regions harbor binding sites for TCF/LEF which cause overexpression of SALL4 by Wnt signaling activation [14]. 

One potential cause of aberrant expression of SALL4 in cancer cells is DNA hypomethylation at CpG islands. The SALL4 CpG islands spanning the first exon and intron regions of the gene are considerably hypomethylated in cancerous cells. Moreover, treatment of B-ALL cells with methylation inhibitors leads to demethylation of SALL4 regulatory regions and its overexpression [26]. On the other hand, methylation-specific PCR (MSP) has validated aberrant hypomethylation of SALL4 promoter in acute myeloid leukemia (AML) and myelodysplastic syndrome (MDS) patients [15,27]. The SALL4-hypomethylated cases showed poor prognosis and lower survival rate than SALL4 normal-methylated ones [27].

Post-transcriptional regulation of SALL4 expression is mediated through regulatory RNAs. miR-33b has a binding site in 3′ untranslated region (UTR) of the SALL4 mRNA and inhibits it in breast cancer cells [28]. Mir-107 also inhibits SALL4 expression in glioma cells [29]. There are several other miRNAs that their regulatory effect on SALL4 has been identified, which have extensively been discussed in our previous review article [30].

From a holistic point of view, these molecules and many other unknown components might enhance SALL4 expression in cooperation with each other. As a result, molecular deviation in any part of this big map may trigger SALL4 oncogenic activation.

## 3. SALL4 Is Aberrantly Activated in Many Types of Cancers

SALL4 overexpression is found in several cancer tissues and hematologic malignancies, suggesting a key role of SALL4 in cancer onset and progression (Figure 1a). Furthermore, the oncogene has been linked to various cellular mechanisms such as proliferation, apoptosis, invasion, and resistance to therapeutics [31] and high SALL4 is associated with poor prognosis and lower survival rate in patients [32]. The Cancer dependency map (Depmap) data, which aim to identify key essential genes in cancer using high-throughput analysis of CRISPR and shRNA, were analyzed by depmap package, R software to assess the essentiality of SALL4 for cancer cells [33]. CIRSPR data showed that SALL4 is an essential driver node with an overall dependency score of −0.13325 among 789 different cancer cell lines (Figure 1b). The score determines the importance of the gene for survival of cancer cells and how its omission is lethal for the cells (a score of 0 shows inessentiality of the gene for survival of the cells, while −1 is the median score of all common essential genes). SALL4 dependency was observed in most of the cancer cell types with the highest dependency score for peripheral nerves system cancers and blood malignancies (Figure 1c, Appendix A). 

### 3.1. SALL4 and Hematologic Malignancies

SALL4 has a critical role in normal hematopoiesis being expressed in CD34+ hematopoietic stem cells and contributes to the maintaining of self-renewal ability and pluripotency of these cells [34]. Immunoprecipitation followed by microarray determined that SALL4 directly regulates key hematopoietic differentiation genes such as CD34, Runt-Related Transcription Factor 1 (RUNX1), HOXA9, and PTEN in CD34+ bone marrow cells [35]. Nevertheless, it is shown that SALL4 is over activated in hematologic malignancies and is linked to worse outcome in patients [4].

Expression of SALL4 in some subtypes of leukemias such as precursor B-cell lymphoblastic leukemia and acute myeloid leukemia demonstrates activation of different molecular machineries in B- and T-cell lymphoblastic leukemias [34]. SALL4 is also highly expressed in anaplastic large cell lymphoma and knock down of the gene has similar effects in restriction of cell-cycle progression and induction of apoptosis [36]. SALL4 is expressed in 75% of chronic myeloid leukemia (CML) patients in blast-crisis, but expressed in just 10% of patients in the chronic phase. CD34+/CD38+ cells that are suggested to be the leukemia-initiating cells (LIC) and promote progression of CML to blast crisis also demonstrate high expression of SALL4 [37]. Shuai et al. showed that SALL4 expression in bone marrow of MDS and AML patients was significantly higher than control group and correlated with expression of MYC and cyclin D1 (CCND1) which are known downstream genes of Wnt/beta-catenin pathway [38]. 

Our findings showed that the level of SALL4 expression is positively correlated with karyotype abnormalities, poor survival, and an increased risk of MDS to AML transformation [39,40]. In another study, we found that SALL4 promotes transformation to AML through downregulation of Fancl (Fanconi anemia, complementation group L). This protein is critical for homologous recombination and DNA damage responses. In addition, it was demonstrated that the anti-apoptotic protein Bcl2 is upregulated in SALL4 transgenic mice, causing more cancer cell survival [41]. In addition, our findings demonstrated that SALL4 binds to Bmi-1 promoter and increases H3-K4 and H3-k79 histone marks, resulting in overexpression of Bmi-1. Bmi-1 drives its oncogenic role through activation of telomerase reverse transcriptase, thus inducing telomerase activity and also promoting cell cycle progression from G1 to S phase [42,43]. Expression of these two oncogenes is correlated in leukemic cells, and both genes are critical for self-renewal ability of the cells [44].

Determination of SALL4 target genes and signaling pathways is critical for development of effective treatment strategies. Chip-chip assay and subsequent gene set enrichment analysis on NB4 leukemic cells demonstrated that SALL4 regulates pathways related to apoptosis, proliferation, inflammation, and cell cycle arrest [45]. SALL4 also binds to HOXA9 promoter in contribution with MLL complex in leukemic cells [46]. SALL4/MLL complex induces histone methylation marks H3K4 and H3K79 and also recruits RNA polymerase II at transcription start site of the HOXA9 gene, resulting in overexpression of this gene [47]. Moreover, SALL4 directly binds to the promoter of retinoic acid receptor α and modulates gene expression through Histone demethylase 1 (LSD1). It is shown that the combination of ATRA and SALL4 inhibitors has a strong restraining effect on AML cells [48].

Various studies report overexpression of SALL4 in drug-resistant leukemic cells and modulation of specific resistance genes by SALL4. Chromatin-immunoprecipitation (ChIP) along with electrophoretic mobility shift assay (EMS) determined biding of SALL4 to the promoter of ATP-binding cassette(ABC)A3, resulting in over expression of this ABC protein and drug-resistance in these cells [46]. Moreover, SALL4 induces expression of another ABC transporter gene named ABCG2 indirectly which both are involved in drug sequestration [49,50]. It is demonstrated that resistant to tyrosine kinase inhibitors such as imatinib, dasatinib, and nilotinib is associated with augmented levels of SALL4 and ABCA3 leading to formation of drug-resistance leukemic cell lines [51].

All together, these findings show that SALL4 is a critical gene for progression and resistance of different hematologic malignancies.

### 3.2. SALL4 and Hepatocellular Carcinoma

Hepatocellular carcinoma (HCC) is the fourth common cancer and the third cause of cancer death in the world. The disease mainly occurs subsequent to liver cirrhosis, hepatitis B or C infection, or nonalcoholic steatohepatitis [52]. Due to ineffective treatments and high mortality rate, HCC attracts high attention, and the generation of novel effective treatments is highly needed.

Various reports show correlation of SALL4 with poor prognosis in patients with HCC [53,54,55]. Indeed, the high expression of SALL4 is shown to be significantly correlated with decreased overall survival in hepatoblastoma samples [56]. In HCC patients, the worst prognosis is seen in lesions with stemness characteristics in which SALL4 overexpression is mainly observed [57]. SALL4 as a cancer stem cell related gene, is suggested as a valuable prognostic biomarker for classification of progenitor-like HCC subtypes [58,59]. Spheroid formation, invasion and drug-resistance all are associated with augmented levels of SALL4 in HCC [58,60,61]. Moreover, as a potential early diagnostic and prognostic biomarker, SALL4 shows a considerably high correlation with parameters of contrast-enhanced ultrasonography in patients with HCC [62]. 

By using two unbiased and precise high throughput assays, we could identify the DNA binding domain of SALL4 and its putative target genes in HCC. The AT-rich motif with a core of WTATB sequence was found to be the consensus binding site of SALL4 to 430 genes. It was found that the chromatin modifiers such as the KDM, Forkhead, BCL, and KLF families are negatively regulated by SALL4, while NKX2-8 was shown to be a activated by this gene [21]. A similar AT-rich motif is reported to be occupied by SALL4, controlling expression of early differentiation genes in embryonic stem cells [63]. In another study, the association of SALL4 immunoreactivity with α-fetoprotein and E-Cadherin is reported [64]. In addition, it is shown that patients with higher expression of SALL4 and α-fetoprotein have a worse prognosis than patients lacking these biomarkers [55]. The prognostic value of SALL4 in primary HCC also is determined in our previous study [58]. By immunohistochemical analysis, we demonstrated SALL4 positivity in 55.6% of the HCC specimens in a Singapore cohort. Microarray expression analysis also showed significant SALL4 upregulation in liver neoplastic lesions compared to non-neoplastic specimens. Functional analysis determined lower proliferation accompanied by increased differentiation in SALL4 downregulated cell lines [58,59]. 

The mechanism(s) of SALL4 activation by hepatitis viruses have been further explored by several groups. One reports on the induction of SALL4 pseudogene 5 (P5) by HBV. The SALL4 P5 then binds to the RNA-binding site of the DNA methyltransferases 1 (DNMT1) which leads to de-methylation and subsequently activation of SALL4 in HCC [17]. This highlights the critical role of none coding RNAs in epigenetic modulations of the gene. Another group also reports the similar CpG island region, which is downstream of SALL4 transcription start site, can be demethylated in HBV-induced HCC. Intriguing, this group also observed SALL4-expression in hepatitis C virus (HCV)-HCC patients and the same demethylation pattern as seen in HBV SALL4-expression HCC cases, even though the detailed mechanism remains unknown. This group further shows that this CpG island is within the binding site of OCT4 and STAT3, and demethylation augments occupancy of these sites by transcription factors and recruitment of chromatin remodeling Brahma-related gene-1 (BRG1), which all result in SALL4 re-activation [65]. 

These findings show that SALL4 reactivation during liver tumorigenesis process represents a key and early event for the cancer onset.

### 3.3. SALL4 and Colorectal Cancer

Colorectal cancer (CRC) is a highly prevalent cancer with high mortality all over the world. Due to lack of early diagnostic biomarkers and insufficient therapeutic approaches, patients struggling with CRC have poor prognoses [66].

High expression of SALL4 is accompanied by worse prognosis in colorectal cancer patients [67]. Wu et al. reported 85.9% sensitivity and 85.7% specificity for SALL4 in serum from CRC patients [68]. Moreover, SALL4 mRNA copy number has been suggested as a probable prognostic and diagnostic biomarker for CRC patients [69]. The number of mRNA at peripheral blood was significantly correlated with the grade of tumor differentiation and tumor invasion. Such correlation has also been reported between the serum and tissue level of SALL4 and lymph node metastasis as well as differentiation degree.

Reports show that SALL4 promotes colorectal cancer malignancy through different molecular mechanisms. MiR-3622a-3p interacts with SALL4 and subsequently suppresses stemness features of CRC cells [70]. SALL4 upregulates Gli1 that is known for its role in cancer development. In addition, Hao et al. showed that SALL4 and β-catenin are positively correlated and interact with each other in CRC cells [67]. 

SALL4 inhibition targets CRC malignant cells through proliferation, invasion, and resistance to therapeutics [71,72]. Silencing of this gene dramatically reduced resistance of CRC cells to fluorouracil and oxaliplatin [71] and lowered the level of anti-apoptotic BCL2 protein in CRC and other types of cancer [72]. 

However, much more study is needed to identify the molecular machinery activated in colon cancer cells post SALL4 over expression and determination of the mechanisms behind resistance in these cells. 

### 3.4. SALL4 and Breast Cancer

Breast cancer is the leading cause of death in females. Due to the insufficiency of current therapeutic approaches mainly in advanced cases of breast cancer, development of new treatment strategies is urgent. The first evidence on the diagnostic value of SALL4 in breast cancer was reported by Kobayashi et al. [73]. Although no correlation between SALL4 and other clinicopathological features was reported, a sensitivity and specificity around 80% was observed for SALL4 expression in breast cancer.

It is demonstrated that the degree of SALL4 expression is significantly related to tumor size and invasion to lymph nodes in breast cancer patients [31]. Yue et al. report correlation of SALL4 expression with tumor size, lymph node stage, and type of breast cancer [74]. High expression of SALL4 also is linked to chemo-resistance of breast cancer cells [75]. 

SALL4 modulates stemness of breast cancer cells through different mechanisms. The basal-like breast cancer cells express high levels of SALL4 through which the alternative splicing of CD44 is modulated [76]. In basal-like breast cancer cells, SALL4 upregulates expression of cell migration related integrin genes ITGA6 and ITGB1 [77]. This network promotes Rho activity and enhances focal adhesion dynamics, causing more cell migration. Moreover, repression of E-cadherin (CDH1) by SALL4 modulates cell dispersion in basal-like cancer types [78]. The oncogene Bmi-1 also is overexpressed by SALL4 in breast cancer cells [28]. Lin et al. determined that miR-33b is a negative regulator of SALL4 which indirectly causes downregulation of Bmi and consequently inhibits the stemness properties of breast cancer cells [28]. On the other hand, SALL4 knockdown caused decrement in the expression of Wnt3a both at RNA and protein level and reduction in sphere formation ability [31]. 

Taken together, SALL4 is a determinant factor in breast cancer cell stemness, proliferation, and invasion, because of which its function as a therapeutic target needs to be further investigated in prospective studies. 

### 3.5. SALL4 and Lung Cancer

Lung cancer has the highest incidence and mortality among different cancers. To increase the survival rate of the patients, early diagnosis of the disease and surgically excising it is important. SALL4 aberrant expression is observed in lung cancer patients with poor survival and it is shown to be a good diagnostic marker with 88% sensitivity and 100% specificity in lung cancer patients [79]. Moreover, Kobayashi et al. proposed SALL4 mRNA level as a diagnostic marker for lung cancer with 85.1% sensitivity and 92.9% specificity [80]. The competence of SALL4 as an immunohistochemical marker for high-grade lung adenocarcinomas with fetal-like morphology is also validated [81]. 

Knock down studies support the importance of SALL4 in progression of lung cancer. SALL4 knock down resulted in down regulation of Cyclin B, Cyclin E, and Cyclin D1 and subsequent cell cycle arrest in the G0/G1 [82]. Moreover, invasion and migration reduced upon SALL4 knock down in vitro and size and weight of the transplanted tumor reduced significantly in the mouse model. On the other hand, shorter disease-free survival was observed in SALL4-high expressing patients demonstrating that SALL4 could be a prognostic marker for NSCLC. [82]. 

EGFR is a transmembrane tyrosine kinase that regulates several signaling cascades. EGFR gene is commonly mutated in non-small-cell lung cancer (NSCLC) patients. Du et al. demonstrated that mutation in EGFR is associated with high expression of SALL4 in NCLC patients [25]. Moreover, they found the correlation of SALL4 expression with stemness of the cancer cells. It is shown that, EGF stimulates SALL4 expression through ERK1/2 signaling pathway. On the other hand, SALL4 in combination with HDAC complex modulates EGFR and IGF1R signaling through suppression of an E3 ubiquitin-protein ligase, CBL-B [83]. 

Similar to other cancer types, overexpression of SALL4 is observed in chemotherapy-resistant lung cancer cells. SALL4 is involved in activation of resistant mechanisms post adjuvant-chemotherapy and knockdown of the gene results in sensitivity of lung cancer cells to platinum-based drugs [61]. In line with this finding, SALL4 is highly expressed in cisplatin-resistant lung cancer cells and knock down of this gene improves response to cisplatin treatment by induction of apoptosis through AKT/mTOR signaling pathways [84]. Taken together, these data suggest that SALL4 targeting may represent a valid therapeutic approach and particularly valuable for EGFR-mutation negative lung cancer cases.

## 4. SALL4 as a Potential Therapeutic Target

In recent years molecular targeted therapy for cancer has attracted a great deal of attention in order to achieve the goals of precision medicine. The functional association of SALL4 with cancer incidence, progression, and metastasis suggests that SALL4-targeting could be effective in eradicating cancer cells [85]. However, transcription factors (TFs) historically have been viewed to be “undrugable”, and multiple issues should be considered in designing efficient therapeutics. For a while, lack of structure knowledge on suitable active or binding sites, localization of the protein in the nucleus, and inaccessibility on the cell surface all make SALL4 as a critical choice for therapeutic drug design [86]. On the other hand, targeting SALL4 can be specific for cancer cells in which SALL4 gene is reactivated and highly expressed, without toxic effects to adjacent normal tissues that do not express SALL4. Motivated by its unique oncofetal properties, our group has designed specific drugs targeting SALL4 which are presented in the following. In addition, along with the advancement in technologies and our growing knowledge of gene regulation, there is an exciting new direction in cancer treatment by targeting TFs, including SALL4. Three different approaches are investigated for targeting SALL4 in cancer cells. One approach targets the functionality of the SALL4 by inhibition of its essential protein–protein interactions. In the second approach, SALL4 targeting drugs are characterized by drug repositioning. The third and more novel approach is modulation of SALL4 protein levels through induction of protein degradation by immunomodulatory drugs (IMiDs), non-IMiD molecular glues, or proteolysis-targeting chimera (PROTAC) technology (Figure 2). The development of degrader-based drugs holds the highest promises in the cancer treatment landscape in near future. 

### 4.1. Approach 1: Target SALL4 Function, the SALL4 Inhibitors

#### 4.1.1. Targeting the Interaction of SALL4 with NuRD Complex

SALL4 interacts with various chromatin modulatory factors such as NuRD complex, DNA methyltransferases, histone methyltransferases, and different histone demethylases. More detailed studies are needed to elucidate the precise interactions between SALL4 and these epigenetic factors. For instance, the interaction of SALL4/HDAC and SALL4/DNMT1 happens through 174 amino acids at N-terminal of the protein and needs to be further mapped as in the cases of SALL4 and RB Binding Protein 4 (RBBP4) interaction [87,88].

NuRD is a chromatin remodeling complex that mainly works as a transcriptional silencer through HDAC and CDH3/4 ATPase subunits and RBBp4. Due to the high impact of SALL4/NuRD interaction on chromatin structure, we tried to halt it through blocking its function. Initially, a 12-AA peptide developed by our group that effectively blocked the interaction between SALL4 and NuRD complex in HCC and acute myeloid leukemia [89]. The peptide treatment can decrease the binding of HDAC to the promoter of PTEN resulting in increment in the expression of PTEN. Further optimization of this peptide led to the discovery of FFW, a pre-clinical prototype, and an SALL4/RBBP4 inhibitor that can induce apoptosis in cancer cells and hindrance of formation of xenograft in mice [88]. We are working to improve its in vivo delivery efficiency currently, and hopefully to translate to the clinic in the near future. 

#### 4.1.2. Inhibition of HDACs

Since SALL4 can recruit NuRD, it is part of this complex with HDAC function. Inhibitor(s) of HDACs (HDACi) can be viewed to inhibit SALL4 function indirectly. As matter of fact, HDACi has been shown to have promising therapeutic effects on SALL4 expressing cancer cells. Connectivity Map (cMap) analysis for determination of possible drugs targeting SALL4 reveals Entinostat (MS275), an HDACi gets the highest score. More experimental studies showed that SALL4-high lung cancer cells are more vulnerable to Entinostat treatment [83]. Surprisingly, additional studies on Entinostat-mediated SALL4 targeting reveals a second drug mechanism. MiRNA sequencing after Entinostat treatment showed that this drug downregulates SALL4 by up-regulation of miR-205 [16]. 

To expand the scope of SALL4 targeted therapy in cancer, we explored a sequential combination therapy approach recently. It has been demonstrated that hypomethylating agents (HMA) which are currently used to treat various types of cancers, activate SALL4 through DNA demethylation as a probable side effect [18]. We started with HMA to prime the SALL4 negative lung cancer cells to become SALL4 positive, therefore, to induce a SALL4-mediated vulnerability in these cells, and then sensitized these cells to the following Entinostat treatment.

### 4.2. Approach 2: Targeting SALL4 Downstream Pathways; Repurposing Oxidative Phosphorylation Drugs to Inhibit SALL4 Positive Cells

Drug repositioning focuses on rediscovery of new targets for current therapeutic agents. This approach accelerates drug development and bypasses the challenging and time-consuming clinical trial steps. In order to identify compounds with inhibitory effect on SALL4, a library of 1597 synthetic and 21,575 natural compounds were tested on SALL4 high expressing cells. The highest score was taken to the oxidative phosphorylation inhibitor Oligomycin. Oligomycin is a macrolide produced by Streptomyces and an inhibitor of adenosine triphosphate (ATP) synthase [90]. This compound demonstrated tumor suppressive role both in vitro and in vivo on SALL4 high cells. In line with this observation, chip-seq data analysis showed binding of SALL4 to near 50% of mitochondrial Oxidative phosphorylation genes. Moreover, oxygen consumption rate and oxidative phosphorylation increased dramatically in SALL4-high cells, all promoting ATP synthesis in these cells [22]. Although glycolysis was previously thought to be the major energy producer of cancer cells, recent findings highlight the critical role of oxidative phosphorylation mechanisms in promotion of tumor progression [89]. 

### 4.3. Approach 3: Modulating SALL4 Abundancy 

#### 4.3.1. Nucleic Acid-Based Therapy

Our growing knowledge on regulatory roles of miRNAs along with development of miRNA specific delivery systems have made miRNA-based therapy as an emerging approach in treatment of cancer. Various miRNAs regulate SALL4 post-transcriptionally among which Let-7 family has been the focus of many investigations. Let-7 binds to SALL4 at 3′ UTR and induces cell differentiation through repression of a cluster of genes. It is shown that Lin28 that is an RNA-binding protein represses Let-7 family members and the Lin28/Let-7 axis is an important player in tumorigenesis [91]. Let-7 overexpression in HCC cells induces cell cycle arrest and apoptosis [92]. Moreover, low Let-7 expression is shown to be a predictor of poor prognosis in lung cancer patients [93]. In line with these findings, miR-98 that is a member of Let-7 family has demonstrated a tumor suppressive role through interaction with SALL4 in HCC, glioma, and ovarian cancer [94,95,96]. There are some other miRNAs that regulate or are regulated by SALL4. MiR-107 directly targets SALL4 and induces apoptosis in glioma cells both in vitro and in vivo [29]. Moreover, miR-296-5p is shown to have a suppressive role on proliferation, migration, and invasion of liver cancer through Brg1/Sall4 axis. Brg1 which enhances expression of SALL4 by binding to its promoter is potentially inhibited by miR-296-5p [97]. MiR-219, miR-16, and miR-103, miR-195, miR-33b, and miR-497 also are proposed to have role in suppression of tumorigenesis via interaction with SALL4 [28,98,99,100]. In a recent systematic review, we comprehensively discussed the interaction between SALL4 and different miRNAs [30]. The inhibitory effect of these miRNAs on SALL4 gene expression can be considered for prospective RNA-based therapy approaches.

There is also a tendency for gene silencing through Small interfering RNAs (siRNA). Knock down of SALL4 by siRNA is shown to induce apoptosis in colorectal and breast cancer cells [72,101]. Ashrafizadeh et al. directly delivered SALL4-siRNA to HCC cells using a lipoprotein-like scaffold which resulted in inhibition of HCC tumor growth in vivo [102]. Moreover, transducing lung cancer cells with siRNA against SALL4 resulted in more sensitivity of the cells to platinum-based drugs [61]. 

The other class of inhibitors that are worth working on are Aptamers. Aptamers are a class of synthesized RNA or DNA oligonucleotides which by their three-dimensional structure specifically bind to target molecules [103]. The beneficial aspects of aptamers as novel drugs are their easy manufacturing, screening, and preformation of chemical modifications against enzyme degradation. The successfulness of RNA-based therapeutics in the market holds the promise for development of efficient aptamers against different SALL4 interactions. 

#### 4.3.2. SALL4 Degraders

Despite their teratogenic effects, IMiDs (immunomodulatory drugs) are applied for treatment of patients with multiple myeloma. These immunomodulatory drugs induce degradation of more than 11 zinc finger proteins including SALL4A [104]. Thalidomide as an ImiD, promotes ubiquitination and subsequent degradation of SALL4A on ZFC2 C2H2 domains by binding to Cereblon (CRBN) in the choline complex E3 ubiquitin ligase (CRL4CRBN) [105]. Based on this function, ImiDs are proposed to be among the new generation of therapeutics which utilize protein degradation machinery to destruct the target protein. Proteolysis-targeting chimaeras (PROTACs) and molecular glues induce the interaction of the target protein with E3 ubiquitin ligases and subsequent degradation of the protein through proteasome [106]. The ImiDs as molecular glues bind simultaneously to the E3 ligases on the one hand and to substrate protein on the other hand, leading to degradation of the target in a proteasome dependent manner. 

PROTACs are the other class of drugs that take advantage of target-specific degradation strategies to omit interested proteins. They are heterobifunctional structures comprising two ligands connected via a linker that one joins to the E3 ligase and the other joins to the protein of interest. The formed ternary complex finally induced Proteasomal degradation of interested protein. However, to the best of our knowledge no PROTAC-based degrader has been designed against SALL4. One potential limiting factors of PROTAC is that it is a larger molecule with a high molecular weight, which may be challenging for the development of an oral drug administration therapy.

Recently, our group developed a novel class of non-IMiD molecular glues that showed an exceptional therapeutic efficacy in xenograft model associated with a very good safety. This new class of therapeutics may represent a new era for molecular glues with the potential to become a disease modifying treatment for intractable cancers. 

## 5. Conclusions

Despite the evident role of SALL4 in tumorigenesis and therapy resistance, no therapeutic targeting SALL4 are currently on the market. The landmark discovery of a new class of molecular glues non-IMiD targeting SALL4 and other oncogenes has the potential to become a real disease modifying treatment and open a novel therapeutic perspective for several intractable cancer conditions.

## Figures and Tables

**Figure 1 cells-11-02601-f001:**
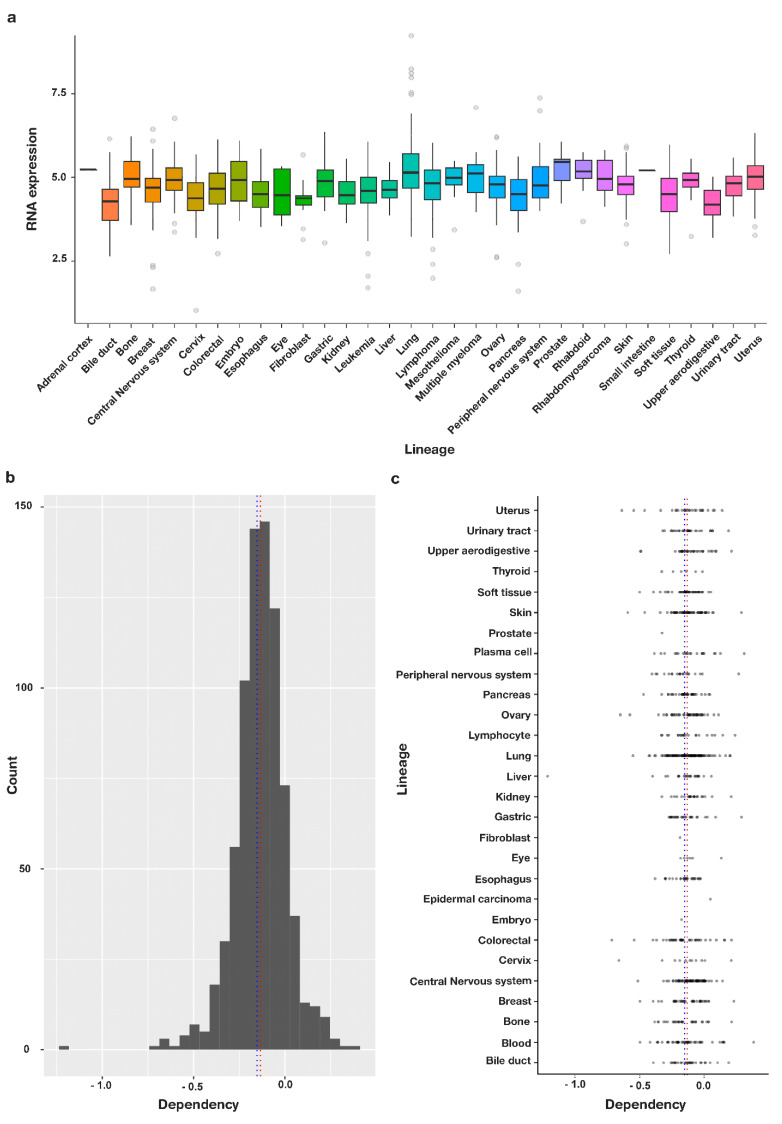
SALL4 is an essential gene for survival of cancer cells. (**a**) Expression pattern of SALL4 across different cancer types. (**b**) The dependency score −0.13325 was calculating by getting average from dependency score of 789 different cell lines for SALL4 (the score determines the importance of the gene for survival of cancer cells and how its elimination is lethal for the cells). (**c**) Categorization of different cell types based on the tissue of origin shows that almost all cancer types have a negative dependency score for SALL4 meaning that they are vulnerable to omission of SALL4.

**Figure 2 cells-11-02601-f002:**
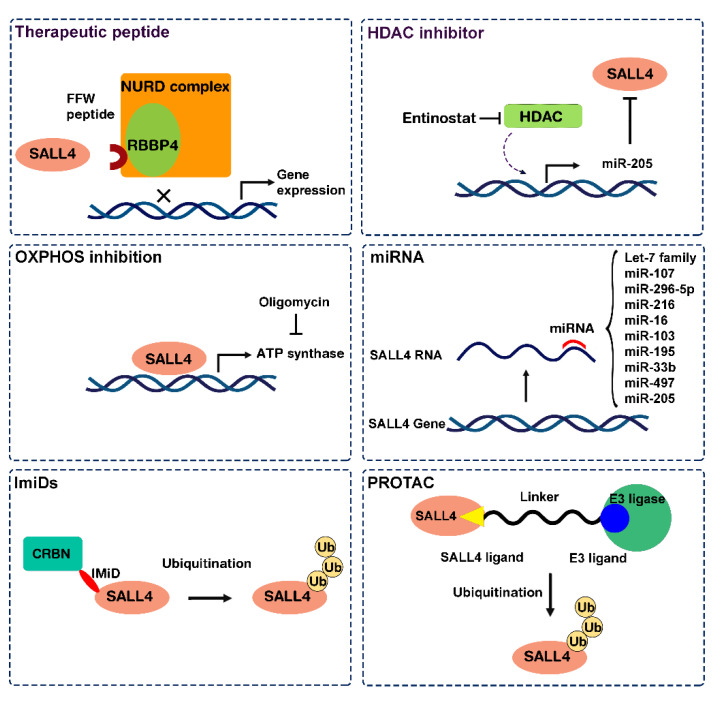
Scheme of the different drugs suggested to target SALL4 in cancer. OXPHOS: Oxidative phosphorylation, IMiDs: Immunomodulatory imide drugs, PROTAC: Proteolysis-targeting chimaeras.

## Data Availability

Not applicable.

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
