# Peer review of "SALL4: An Intriguing Therapeutic Target in Cancer Treatment"

_cells, 2022, doi:10.3390/cells11162601_

Round 1

Reviewer 1 Report

The review entitled “SALL4: an intriguing therapeutic target in cancer treatment” is an interesting review. The authors compiled extensive information about SALL4. But there are some minor points that should be edited by the authors.Below you can find them:

 *the authors should write the full name of SALL4. Also they should browse through the manuscript and write full names of other genes, transcription factors and signalling pathways.

*what do the authors mean by HR on page 1, lines 29 and 30? They should write the full name of HR.

*In figure 1a “breast” should be corrected. There is not bar for adrenal cortex and small intestine, why?

Author Response

Response to Reviewer 2 Comments

The review entitled “SALL4: an intriguing therapeutic target in cancer treatment” is an interesting review. The authors compiled extensive information about SALL4. But there are some minor points that should be edited by the authors. Below you can find them:

Point 1: The authors should write the full name of SALL4. Also, they should browse through the manuscript and write full names of other genes, transcription factors, and signaling pathways

Response: The full name of the genes and the signaling pathway was added to the text in the revised manuscript.

Point 2: what do the authors mean by HR on page 1, lines 29 and 30? They should write the full name of HR.

Response: HR stands for hazard ratio which is corrected in the text according to the reviewer’s wish.

Point 3: In figure 1a “breast” should be corrected. There is no bar for the adrenal cortex and small intestine, why?

Response: We have corrected “Breast” in figure 1a. The CRISPR knockdown data was retrieved from the depmap database that contains only one small intestine and one adrenal cortex cell line. 

Reviewer 2 Report

The manuscript by Moein and colleagues titled “SALL4: an intriguing therapeutic target in cancer treatment” is a thorough review of SALL4 function discussing its correlations with various solid tumors concluding with a couple of different ways to therapeutically target SALL4.  Overall, the review is well written and referenced.  There are a few minor concerns noted below.

1 - The Introduction (1) and Molecular Mechanisms...(2) sections would benefit from an overall model of SALL4 activation and where it affects the pathways and genes listed in the introductory paragraphs More detail is needed to paint a picture of SALL4’s general functions in cell biology to properly understand how SALL4 function in cancer is different.  Currently the most of these sections read like lists without enough context to understand why they are important to the overall model of SLL4 function in both normal cell function and cancer.

2 - Figure 1 needs more detail provided to understand how the RNA expression and Dependency were calculated.  Also, the information identifying the data set(s) used should be provided. 

3 - It is not clear that a Depmap score of -0.1 is meaningful for a gene described as “the 11th most vital gene for cancer cell survival.”  It is also not clear how this Depmap data indicates SALL4 is a “driver node.”  Further confounding this description is the discussion provided in Section 4 which seems to indicate that targeting SALL4 is a viable option as it will not have an effect on adjacent normal tissue.  A more complete and clear explanation is required to clear this confusion.

Author Response

Response to Reviewer 3 Comments

The manuscript by Moein and colleagues titled “SALL4: an intriguing therapeutic target in cancer treatment” is a thorough review of SALL4 function discussing its correlations with various solid tumors concluding with a couple of different ways to therapeutically target SALL4.  Overall, the review is well-written and referenced.  There are a few minor concerns noted below.

Point 1: The Introduction (1) and Molecular Mechanisms...(2) sections would benefit from an overall model of SALL4 activation and where it affects the pathways and genes listed in the introductory paragraphs More detail is needed to paint a picture of SALL4’s general functions in cell biology to properly understand how SALL4 function in cancer is different.  Currently, most of these sections read like lists without enough context to understand why they are important to the overall model of SLL4 function in both normal cell function and cancer.

 Response: This reviewer suggests more details on the general function of SALL4. In this regard, a paragraph about SALL4 function in normal development was added to the introduction of the manuscript. 

Point 2: Figure 1 needs more detail provided to understand how the RNA expression and Dependency were calculated.  Also, the information identifying the data set(s) used should be provided. 

Response : This respected reviewer is concerned about the data used for the preparation of figure 1 and the procedure to reach these figures. In order to clarify how the data is analyzed supplementary data 1 is added to the manuscript which contains the codes used in R software for analyzing the data and also the datasets of the depmap project that were applied in this study.

Point 3: It is not clear that a Depmap score of -0.1 is meaningful for a gene described as “the 11th most vital gene for cancer cell survival.”  It is also not clear how this Depmap data indicates SALL4 is a “driver node

Response: As is mentioned in the text “The score determines the importance of the gene for the survival of cancer cells and how its omission is lethal for the cells (a score of 0 shows the inessentiality of the gene for survival of the cells, while -1 is the median score of all common essential genes).” According to the CRISPR data for SALL4 gene that is provided in the depmap database (https://depmap.org/portal/gene/SALL4?tab=overview), this protein is strongly selective meaning that this genes dependency is at least 100 times more likely to have been sampled from a skewed distribution than a normal distribution.

Point 4:  Further confounding this description is the discussion provided in Section 4 which seems to indicate that targeting SALL4 is a viable option as it will not have an effect on adjacent normal tissue.  A more complete and clear explanation is required to clear this confusion.

Response : As SALL4 is not expressed in normal human tissues, anti-SALL4 treatment would be a safe approach for specific targeting of cancer cells.